# Effectiveness and Safety of Fenofibrate in Routine Treatment of Patients with Hypertriglyceridemia and Metabolic Syndrome

**DOI:** 10.3390/diseases11040140

**Published:** 2023-10-13

**Authors:** Marat V. Ezhov, Gregory P. Arutyunov

**Affiliations:** 1Federal State Budgetary Institution National Medical Research Center of Cardiology, Moscow 121552, Russia; 2Department of Internal Medicine, Pirogov Russian National Research Medical University, Moscow 117997, Russia; arut@ossn.ru

**Keywords:** hypertriglyceridemia, non-HDL-cholesterol, triglyceride, C-reactive protein, fenofibrate

## Abstract

**Background:** Multiple trials have demonstrated the efficacy of fenofibrate for the management of dyslipidemia. Real-world evidence may provide important insights into the effectiveness and safety of fenofibrate in patients with metabolic syndrome and elevated triglyceride (TG) levels, but such evidence is currently scarce. Materials and Methods: A non-interventional study was conducted among routine healthcare providers. Patients with TG levels of >2.3 mmol/L on stable statin therapy starting fenofibrate treatment were enrolled. Data on medical history, fenofibrate treatment, change in lipid levels, and C-reactive protein (CRP) were collected from medical records every 3 months for 6 to 7 months of observation. Results: Overall, 988 patients receiving fenofibrate were enrolled (median age [95% CI] 60 [26.0–86.0] years), and 46.4% of the participants were females. Most patients had concomitant cardiovascular disease. A baseline TG level of 3.6 ± 1.5 mmol/L was reduced by 50.1% to 1.7 ± 0.58 mmol/L at 6 months of treatment (*p* < 0.001). Baseline non-high-density lipoprotein cholesterol (non-HDL-C) levels decreased by 33.7% at 6 months. Total cholesterol and low-density lipoprotein levels by the end of follow-up were reduced by 24.7 and 25.5% (*p* < 0.001 for both). C-reactive protein level decreased more than 39% from baseline. Conclusions: Fenofibrate in a real-world setting significantly reduced TG, LDL-C, and non-HDL-C levels. In addition, a C-reactive protein level reduction of 39% was achieved.

## 1. Introduction

Hypertriglyceridemia and metabolic syndrome are interconnected conditions that contribute to a heightened risk of cardiovascular diseases and type 2 diabetes [1]. The worldwide increasing prevalence of hypertriglyceridemia and metabolic syndrome [2] necessitates further studies on effective treatment options to reduce associated morbidity and mortality. The World Health Organization (WHO) defines metabolic syndrome as the presence of insulin resistance, i.e., type 2 diabetes mellitus (T2DM) or impaired fasting glucose (>100 mg/dL or >5.6 mmol/L), accompanied by at least two of the following: abdominal obesity, elevated triglycerides (TGs) and/or low high-density lipoprotein cholesterol (HDL-C), blood pressure above normal, and microalbuminuria [3]. Globally, the prevalence of metabolic syndrome is estimated to affect around 20–25% of the adult population [4]. In Europe, the prevalence ranges from 10 to 40%, depending on the specific population and diagnostic criteria [5]. In Russia, metabolic syndrome affects approximately 25.8% of the adult population, with regional variations [6].

Fibrates have demonstrated efficacy in lowering TG levels and improving lipid profiles [7]. Fenofibrate acts as a peroxisome proliferator-activated receptor alpha (PPAR-α) agonist regulating the expression of genes involved in lipid metabolism [8,9]. The fenofibrate-mediated activation of PPAR-α leads to various consequences, including increased lipolysis via lipoprotein lipase activation and apolipoprotein C-III suppression, increased fatty acid uptake and acyl-coenzyme A synthase induction in the liver, and stimulation of apolipoprotein A-I and A-II synthesis [8,9,10,11,12]. Clinical trials have demonstrated fenofibrate’s ability to reduce TG levels by up to 50% and increase HDL-C by up to 20% [7].

Hypertriglyceridemia is of particular concern because elevated TG levels are independently associated with an increased risk of cardiovascular events [13]. TGs contribute to atherogenesis by promoting the formation of small low-density lipoprotein cholesterol (LDL-C) particles and impairing the function of HDL cholesterol, which ultimately lead to the development of atherosclerosis [14]. Moreover, hypertriglyceridemia is also associated with other cardiovascular risk factors, such as insulin resistance, inflammation, and endothelial dysfunction [15].

While LDL-C’s role is well understood by physicians and proper attention is being paid to its evaluation and management, the role of TGs and non-high-density lipoprotein cholesterol (non-HDL-C) is still underestimated, and commonly, these parameters are not routinely evaluated. It is important to note that the 2021 ESC Guidelines on cardiovascular disease prevention highlight these parameters’ role in cardiovascular morbidity and mortality, and therefore, a new risk-assessment scale, SCORE 2, was introduced. Moreover, this scale includes non-HDL-C as a contributing factor for 10-year mortality risk calculation [16].

Thus, proper attention needs to be paid to physicians’ awareness and education as well as to timely diagnosis and proper management of these lipid parameters.

One of the landmark randomized clinical trials assessing the efficacy of fenofibrate in patients with hypertriglyceridemia and metabolic syndrome is the Fenofibrate Intervention and Event Lowering in Diabetes (FIELD) study. The FIELD study included 9795 participants with type 2 diabetes and demonstrated a non-significant 11% reduction in the primary composite endpoint of non-fatal myocardial infarction and coronary heart disease death. At the same time, the risk of non-fatal myocardial infarction was reduced by 24% [17].

Another important trial is the Action to Control Cardiovascular Risk in Diabetes (ACCORD) Lipid trial [18]. This study included 5518 participants with type 2 diabetes and high cardiovascular risk and investigated the combination of fenofibrate and simvastatin compared to simvastatin alone. A subgroup analysis revealed that participants with high TG and low HDL-C levels experienced a significant reduction in cardiovascular events.

These studies focused on specific populations, such as patients with type 2 diabetes, which may not be generalizable to the broader population with hypertriglyceridemia and metabolic syndrome. Additionally, the FIELD and ACCORD Lipid trials mainly assessed fenofibrate’s impact on cardiovascular outcomes, with less emphasis on the drug’s effect on insulin resistance, inflammation, and combination therapy potential. It is also noteworthy that there is a particular lack of real-world evidence on fibrates’ effectiveness and safety [19].

The Epidemiology of Cardiovascular Risk Factors and Diseases in Regions of the Russian Federation (ESSE-RF) is a large-scale study conducted in Russia that provides valuable data on the prevalence, awareness, treatment, and control of cardiovascular risk factors and diseases [20]. Hypertriglyceridemia was found to be quite common among the Russian population, with every third man (30.1%) and every fifth woman (21.4%) affected [21]. This study offers important insights into the cardiovascular health of the Russian population, but it does not specifically focus on the use of fenofibrate in patients with hypertriglyceridemia and metabolic syndrome.

By conducting this non-interventional study in a real-world setting, we aim to provide more robust evidence for the use of fenofibrate in real-world clinical practice in Russia, potentially improving the management and outcomes for patients with hypertriglyceridemia and metabolic syndrome.

## 2. Materials and Methods

### 2.1. Study Setting

The study was conducted in clinics/facilities involved in the routine healthcare of patients with hypertriglyceridemia and metabolic syndrome. Each healthcare organization could be represented by one principal investigator to avoid duplicate inclusion of one patient. After preliminary assessment of 152 clinical centers, 82 investigators from 35 regions of the Russian Federation agreed to participate in the study.

### 2.2. Participants

The study included adult (≥18 years old) male or female patients that had TG levels above 2.3 mmol/L on stable therapy with statins. Patients were required to have not received fenofibrate treatment for at least 3 months before inclusion in the study and prescribed with current fibrate treatment in routine practice ≤3 days before inclusion in the study.

For safety reasons, patients who were pregnant or breastfeeding females and/or who had known statin intolerance, type 1 diabetes, chronic kidney disease (CKD), liver failure, and/or pancreatitis were not involved in the study.

### 2.3. Study Design

The current study was conducted as an observational study. All participants signed informed consent forms prior to any study-related procedures. Fenofibrate was prescribed at 0 to 3 days prior to the study. The only available fenofibrate formulations in the Russian Federation are 145 mg film-coated tablets. The decision to prescribe fenofibrate was not made based on this protocol but in accordance with routine clinical practice. Considering the non-interventional nature of the study design, data collection and entry (hereinafter referred to as “visits”) were planned for timelines representing a physician visit schedule of every 3 months for 6 to 7 months of observation recommended in routine healthcare in Russia. To avoid missing data, every follow-up visit entry was allowed 30 days before/after the assumed visit date: 3 months for Visit 2 and 6 months for Visit 3. Therefore, follow-up duration in the study could vary from 5 to 7 months.

In addition to visits, every 30 days, an investigator contacted participants by phone to evaluate their safety and treatment compliance.

### 2.4. Ethical Approval

The study was organized in accordance with the principles of Good Clinical Practice, the Declaration of Helsinki, and applicable local regulations. Every patient was informed about all aspects of the study and had an opportunity to answer any study-related question before signing the informed consent form.

The study was approved by the Independent Interdisciplinary Committee for Ethical Expertise of Clinical Trials (Moscow, Russia), protocol #15 from 29 September 2020.

The study was registered at ClinicalTrials.gov (accessed on 9 October 2023) (NCT04650152).

### 2.5. Study Procedures

The study used both primary and secondary data collection: case report form filling was mostly based on patients’ medical records obtained from their physicians. At every visit, an investigator evaluated vital signs, weight and height, waist circumference, treatment status, concomitant diseases and conditions, results of blood or serum tests for lipids, glucose tolerance testing, and CRP levels.

Treatment adherence was evaluated by a self-reported statement of treatment compliance during the visits and phone calls with investigators.

The only study-related procedure was the completion of a RAND 36-Item Health Survey 1.0 form. The use of a quality-of-life questionnaire in the study is not considered as an intervention according to national regulations.

Information about ADRs was collected based on reporting by an investigator. ADRs were coded according to the Medical Dictionary for Regulatory Activities (MedDRA).

### 2.6. Endpoints

TG level at Visit 3 (5–7 months after inclusion) was used as a primary endpoint. TG level changes were also assessed in subgroups of special interest:-T2DM;-Impaired glucose tolerance at inclusion;-Concomitant cardiovascular (CV) disease;-Perimenopausal women;-Postmenopausal women.

Secondary endpoints comprised changes in HDL-C, LDL-C, total cholesterol, non-HDL-C, and CRP levels.

### 2.7. Statistical Methods

Considering the non-interventional nature of this study and the absence of a control group and statistical hypothesis, descriptive statistical methods were used. To analyze parameters’ changes, the mixed model repeated measures (MMRM) method was used. MMRM included baseline values as covariates. Baseline characteristics and effectiveness for primary and secondary endpoints were analyzed in the FAS population, which included all patients who were eligible for inclusion and had data about treatment initiation. The per-protocol (PP) population comprised patients who finished the study without any significant protocol deviations and had data for an effectiveness assessment. The safety population included all enrolled patients who received at least one dose of fenofibrate.

## 3. Results

### 3.1. Study Population

Figure 1 presents a flow chart of the patients in this study. Initially, 1000 participants were enrolled in the study; however, the data of 12 participants were excluded from the analysis due to the absence of sufficient data on fenofibrate use during the follow-up.

The baseline population characteristics, medical history, and concomitant medications for the full analysis set (FAS) are presented in Table 1.

Hypertension was reported in 80% of patients, and T2D was registered in more than 25% of participants. About 33% of the patients (n = 307) had documented coronary heart disease, and 10% of patients had a history of myocardial infarction.

All patients received statin treatment as per the inclusion criteria. About 80% of patients were receiving either angiotensin-converting enzyme inhibitor or angiotensin receptor blockers. Almost 50% of participants received either antiplatelets or anticoagulants. Treatment for diabetes was reported for 21.3% of all patients or for 210 out of 265 diabetics.

### 3.2. Fenofibrate Treatment Compliance

Compliance was evaluated monthly both at face-to-face visits and via phone calls. Most patients reported medication adherence at both Visits 2 and 3 with compliance rates of 99.2 and 97.8%, respectively.

### 3.3. Triglyceride Level Change

#### 3.3.1. Overall Cohort

Figure 2 presents changes in TG level during the study period. Mean ± standard deviation (SD) baseline TG level was 3.6 ± 1.5 mmol/L. After 3 months of treatment, the mean TG level decreased to 2.2 ± 1.0 mmol/L, indicating a 36.6% change. At Visit 3, the mean TG concentration was 1.7 ± 0.58 mmol/L, with a reduction from the baseline by 50.1% (*p* < 0.0001 for Visits 2 and 3 compared to Visit 1).

#### 3.3.2. Target Subgroups

Figure 3 presents changes in the TG levels among the main preplanned subgroups of interest, including patients with metabolic syndrome, overweightness and obesity, concomitant T2DM, and cardiovascular disease and impaired glucose tolerance as well as female patients in the peri-/postmenopausal period.

TG changes were consistent with the results from the overall population for all subgroups, with some minor and clinically insignificant differences. Therefore, the characteristics of the selected subgroups did not impact the effectiveness of fenofibrate in terms of TG level reduction.

### 3.4. Non-High-Density Lipoprotein Cholesterol

Non-HDL-C levels decreased significantly in patients on fenofibrate by 22.5% at 3 months and 33.7% at 6 months (Figure 4 (*p* < 0.001 for Visits 2 and 3 compared to Visit 1).

### 3.5. Other Laboratory Markers

Figure 5 and Table A1 show significant changes in total cholesterol, LDL-C, HDL-C, and C-reactive protein (CRP) levels in the FAS population.

Total cholesterol and LDL-C levels decreased by one-quarter from baseline (24.7 and 25.5%, respectively, *p* < 0.001 for both), whereas levels of HDL-C increased by 23% (*p* < 0.001). C-reactive protein levels decreased from baseline by 30% at Visit 2 and by 40% at Visit 3 (*p* < 0.001 for both).

### 3.6. Safety

Investigators reported only one adverse drug reaction (ADR): a 70-year-old man receiving concomitant treatment with warfarin had international normalized ratio (INR) lability.

## 4. Discussion

### 4.1. Effectiveness Results

#### 4.1.1. Fenofibrate Studies

##### Impact on Triglycerides

The results of this study demonstrate a significant reduction in TG levels among patients treated with fenofibrate who were previously treated with statins. In the overall cohort, there was a 36.6% reduction in TG levels after approximately 3 months of treatment, with a further decrease to 50.1% at 6 months. Similar reductions were observed among various subgroups, indicating that the effectiveness of fenofibrate in lowering TG levels was consistent across different patient populations.

These findings are consistent with results from other studies that have evaluated the efficacy of fenofibrate in reducing TG levels. The ACCORD Lipid trial also reported a significant decrease in TG levels among patients with type 2 diabetes receiving fenofibrate added to simvastatin compared to those receiving simvastatin alone [18]. Notably, long-term treatment with fenofibrate in the ACCORD Lipid clinical trial led to the maintenance of or decrease in TG levels in the post-trial observation period, indicating a legacy effect of fibrate treatment [22]. Park et al. reported that the addition of fenofibrate to stable statin treatment in patients with controlled LDL-C and elevated TG levels resulted in a significant TG decrease [23]. Moreover, a meta-analysis that included 18 RCTs showed a significant reduction in TG levels among patients treated with fenofibrate, with an average reduction of 38.7% [7].

Observational studies have reported similar results. Kim et al. reported that the addition of fenofibrate to statin in a Korean real-world propensity matched cohort improved reduction in serum TG with combined treatment [24]. In the retrospective study of Kayıkçıoğlu et al. (2020), an analysis of a population with long-term follow-up indicated that a significant decrease in TG level established after 1 year of fibrate treatment was improved by extending the treatment period up to 15 years [25]. These findings are in line with the 36.6% reduction observed in our study after 3 months of treatment. Earlier, Choi et al. reported that the addition of fenofibrate to statin improved lipid profiles including TGs [26].

Data from the systematic review and meta-analysis performed by Rodriguez-Gutierrez et al. (2023), focusing on the efficacy of fibrates in T2DM patients, indicated a mean decrease of 17.8% from the baseline [27]. Our analysis of a real-world population indicates a more than two-fold decrease in TG levels among individuals with T2DM by the end of the follow-up.

An analysis of approved clinical trials indicates that patients with metabolic syndrome may benefit more from fibrate treatment due to higher baseline TG levels [28]. In our study, the use of fenofibrate for metabolic syndrome patients resulted in a two-fold TG level decrease in 5–7 months of treatment.

Menopausal transition may be associated with gradual deterioration in lipid profiles, including TGs [29,30]. The postmenopausal population from our study had TG levels comparable to the overall sample with effectiveness no different from the overall population.

Treatment adherence was evaluated by a self-reported statement of treatment compliance during the visits and phone calls with investigators. More than 95% of patients reported receiving medication. Kiortsis et al. reported that compliance evaluated objectively by biomarker change was associated with a better TG lowering effect of fenofibrate [31]. However, compliance monitoring in routine practice might be subject to reporting bias not only for medication, but also for lifestyle changes. Different tools, including digital solutions, might improve treatment-adherence monitoring [32].

##### Non-High-Density Lipoprotein Cholesterol Change

In addition to the efficacy in lowering TG levels, our study also found a significant reduction in non-HDL cholesterol levels during fenofibrate treatment, which is consistent with the findings of other studies, such as the FIELD [17] and ACCORD Lipid [18] trials.

Non-HDL-C level is an established factor related to cardiovascular risk [33]. In our study, fenofibrate added to statin treatment in a real-world cohort resulted in significant improvement of non-HDL-C levels. Similar results were observed in a randomized controlled FIELD study [34]. The results of a SAFARI trial indicated additional reduction in non-HDL-C with fenofibrate adjunct to simvastatin compared to statin monotherapy [35]. In a recent study, Ihm et al. (2020) reported that, in patients with stable LDL-C and elevated TG levels, adding fenofibrate to pitavastatin led to a better reduction in non-HDL-C levels compared to pitavastatin monotherapy (treatment difference of 12.45% at 8 weeks in favor of combination treatment) [36].

##### Impact on Inflammatory Status

In our study, adding fenofibrate to statin treatment led to a 38.9% reduction in serum CRP levels. Ihm et al., in the abovementioned clinical study, reported a reduction in CRP with a pitavastatin/fenofibrate combination [36]. There are several mechanisms of inflammatory process involvement in atherosclerotic plaque formation and destabilization, including macrophages, neutrophils, lymphocytes, interleukins, and chemokines [37]. In particular, CRP levels detected with high-sensitivity testing were recognized as independent predictors of cardiovascular outcomes [38]. Therefore, fenofibrate might contribute to risk modification not only through its impact on the lipid profile, but also by altering inflammation, which requires further elucidation.

#### 4.1.2. Other Fibrates

Several studies of other fibrates concerning dyslipidemia treatment have been published in recent years. Machado-Duque et al. (2020) have reported a real-world study of fenofibric acid effectiveness in patients with mixed dyslipidemia. Treatment for ≥3 months resulted in a 64.2% baseline TG level reduction [39]. Regarding bezafibrate, the results from a recent randomized study published by Nakamura et al. (2023) indicated a 34.7% decrease in baseline TG levels at 6 months in patients with hypertriglyceridemia [40].

Pemafibrate is the latest class fibrate medication that was expected to reach the market in the past few years. A PROMINENT study demonstrated promising results with a TG reduction of 26.2% after 4 months of treatment [41]. Despite these results, treatment with pemafibrate was associated with cases of significant increase in low-density lipoprotein levels [41], which were also observed in a network meta-analysis comparing pemafibrate efficacy to fenofibrate [42].

### 4.2. Safety

Regarding safety, our study reported only one mild and non-serious ADR in a single patient receiving concomitant treatment with warfarin. This finding suggests that fenofibrate is generally well tolerated. However, it is important to consider the potential underreporting of ADRs in the Russian Federation and Eastern European countries, as highlighted in this study.

The safety profile of fenofibrate appears favorable, but further research should be conducted to confirm these findings, particularly in regions where ADR reporting may be less comprehensive. In the FIELD study, the discontinuation rate was 12.5% for fenofibrate-treated patients [17] while in the ACCORD Lipid trial, the discontinuation rate for patients receiving fenofibrate plus simvastatin was 13.5% [18].

A routine population analysis performed by Kayıkçıoğlu et al. (2020), covering up to 15 years of follow-up in 187 fibrate (mostly fenofibrate)-treated patients, indicates ADRs reported in eight patients, with five of the eight ADRs attributed to myalgia or liver transaminase increase. It is noteworthy that half of the patients analyzed were treated with a fibrate/statin combination [25].

Regarding other drugs from the fibrate class, Machado-Duque et al. (2020) reported that 1% of patients treated with a rosuvastatin/fenofibric acid combination had ADRs related to treatment in a real-world setting [39]. In a randomized study of pemafibrate and bezafibrate, adverse events were reported in 43 and 37% of patients, respectively [40].

When analyzing real-world safety data, it should be taken into account that, currently, there is a significant gap between ADR reporting in routine practice and clinical trials [43].

### 4.3. Study Strengths and Limitations

Real-world data collection and analysis provide numerous advantages for both investigators and routine practice. The major advantage of the real-world approach is the analysis of the medication effect in the wider population while clinical trial samples are usually strictly narrowed by the inclusion criteria. Moreover, real-world studies may provide larger samples for an analysis compared to clinical trials [44,45]. It should also be noted that our study is the first attempt to analyze the real-world effectiveness of fibrates in the Russian Federation population.

In addition to the general limitations of non-interventional studies, this study has some specific limitations.

Ad hoc analysis of patients’ compliance with diagnostic criteria: It is important to note that an ad hoc analysis of patients’ compliance with the accepted diagnostic criteria for metabolic syndrome will be published later. This analysis may reveal additional insights into the patient population and the potential impact of diagnostic practices on the study outcomes. Despite these limitations, this study provides valuable information on the effectiveness and safety of fenofibrate in a real-world setting, adding to the existing body of evidence from randomized clinical trials and observational studies.

## 5. Conclusions

Fenofibrate in a real-world setting significantly reduced TG, LDL-C, and non-HDL-C levels. In addition, a C-reactive protein level reduction of 39% was achieved. Our study adds to the growing body of evidence supporting the effectiveness of fenofibrate in better-controlled TG and non-HDL-C levels among patients with metabolic syndrome.

## Figures and Tables

**Figure 1 diseases-11-00140-f001:**
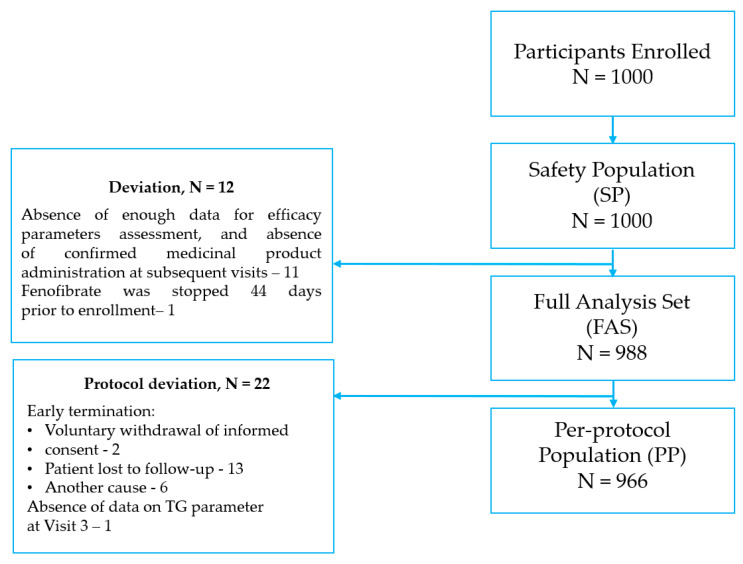
Patient flow chart.

**Figure 2 diseases-11-00140-f002:**
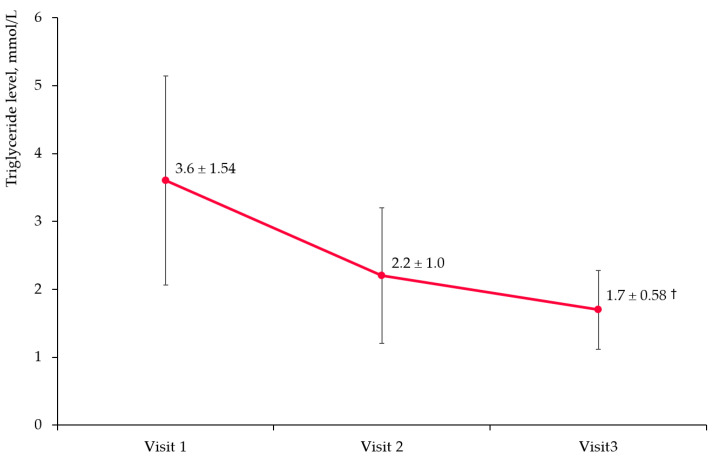
Triglyceride level changes in overall cohort. Notes: The dots represent mean values and the error bars are standard deviations; †—*p* < 0.0001 compared to baseline.

**Figure 3 diseases-11-00140-f003:**
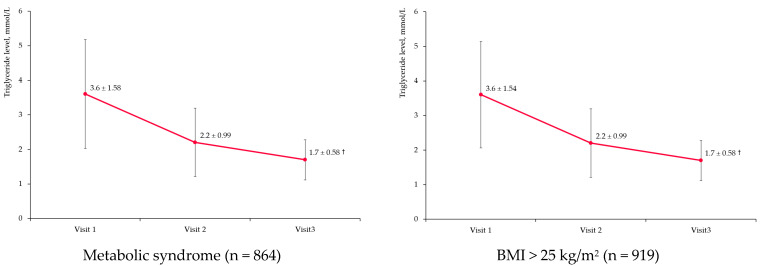
Triglyceride level change in target subgroups. Notes: BMI—body mass index; T2DM—type 2 diabetes mellitus; the dots represent mean values, and the error bars are standard deviations. † *p* < 0.001 compared to Visit 1.

**Figure 4 diseases-11-00140-f004:**
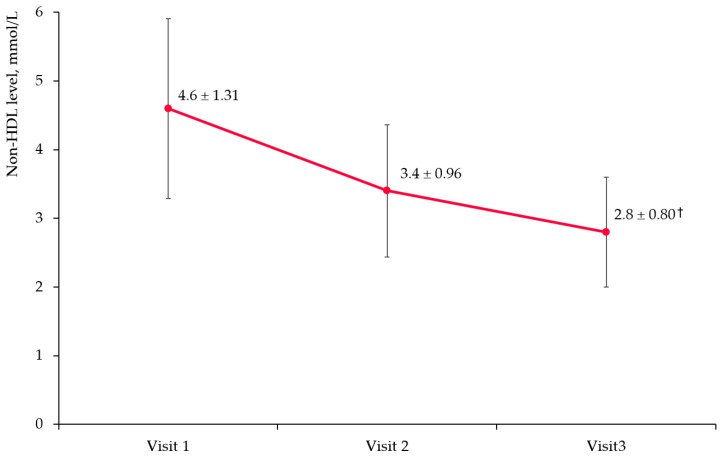
Non-HDL cholesterol level changes in overall cohort. Notes: HDL—high-density lipoprotein; †—*p* < 0.0001 compared to baseline.

**Figure 5 diseases-11-00140-f005:**
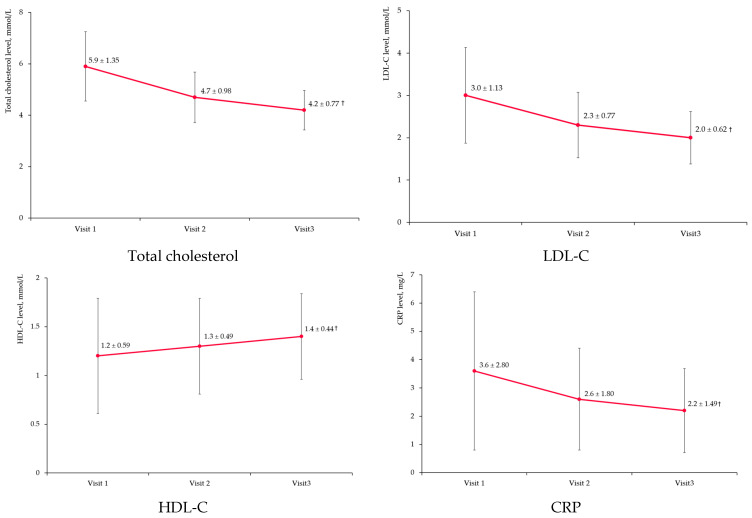
Other lipid profile parameters and C-reactive protein level changes. Notes: LDL-C—low-density lipoprotein cholesterol; HDL-C—high-density lipoprotein cholesterol; CRP—C-reactive protein; the dots represent mean value and the error bars are standard deviations; †—*p* < 0.001 compared to baseline.

**Table 1 diseases-11-00140-t001:** Baseline characteristics of study population (n = 988).

Parameter	Value
Age, years, mean (SD)	58.3 (10.11)
Female sex, n (%)	458 (46.4)
BMI, kg/m^2^, mean (SD)	31.2 (4.67)
Race, n (%)	
Caucasian	972 (98.4)
Asian	16 (1.6)
Smoking status, n (%)	
Smoker	797 (80.7)
Non-smoker	191 (19.3)
Menopausal status, n (%)	
Perimenopause	45 (9.8)
Menopause	361 (78.8)
Not applicable	52 (11.4)
Concomitant disease	
Hypertension	784 (79.36)
Type 2 diabetes	265 (26.82)
Coronary heart disease	307 (31.07)
Myocardial infarction	100 (10.12)
Group of concomitant medications	
Statins	963 (97.47)
Renin-angiotensin blockers	799 (80.87)
Beta blockers	565 (57.19)
Antithrombotics	467 (47.27)
Diuretics	275 (27.83)
Calcium channel blockers	263 (26.62)
Antidiabetics	210 (21.26)

Notes: BMI—body mass index.

## Data Availability

The data presented in this study are available on request from the corresponding author. The data are not publicly available due to national regulations.

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
