# Peer review of "Effectiveness and Safety of Fenofibrate in Routine Treatment of Patients with Hypertriglyceridemia and Metabolic Syndrome"

_diseases, 2023, doi:10.3390/diseases11040140_

Round 1
Reviewer 1 Report
Dear Authors,
I read with interest Your manuscript entitled "Effectiveness of fenofibrate in routine treatment of hypertriglyceridemia".
Here are my comments and suggestions.
1) Title :
You wrote "Effectiveness of fenofibrate". However, You wrote about effectiveness and safety of fenofibrate.
2) Introduction:
What definition of metabolic syndrome did You refer to ? Please, specify.
The mechanism of fenofibrate action should be better specified and discussed. References should be modified accordingly.
3) Study design:
Dosage of fenofibrate ? It is not clear. Is there only one dosage of fenofibrate in the Russian Federation ?
4) 3.3.2 Target subgroups:
You wrote that "TG changes in the prespecified subgroups were consistents with results from the overall cohort" (please, see lines 193 and 194). You defined the subgroups of special interest as follows: a) type 2 diabetes mellitus (T2DM); b) impaired glucose tolerance at inclusion; c) concomitant cardiovascular (CV) disease; d) perimenopausal women; e) postmenopausal women. Figures relating these results were not present. This is a major comment.
5) Discussion:
should be modified accordingly.
The Safety Section (4.2) should be re-written.
Strenghts of Your study?
Lines from 292 to 298 should be removed, because they greatly reduced the scientific soundness of Your manuscript.
6) Extensive editing of English language is required by an English native speaker, and some typing errors must be removed.
7) References should be updated.
Extensive editing of English language is required.
I suggest a stylistic revision by a English native speaker.
Author Response
Response to Reviewer Comments and Suggestions
Dear Reviewer,
Thank you for your insightful comments and suggestions on our manuscript titled "Effectiveness of fenofibrate in routine treatment of hypertriglyceridemia". We value your feedback and have taken steps to address each of your concerns. Please find below our responses to your comments:
1) Title:
Comments: "Effectiveness of fenofibrate". However, you wrote about effectiveness and safety of fenofibrate.
Response 1: Thank you for pointing this out. We have revised the title to appropriately capture the content of our manuscript.
[Effectiveness and safety of fenofibrate in routine treatment
of patients with hypertriglyceridemia and metabolic syndrome]
2) Introduction:
Comments: Definition of metabolic syndrome and mechanism of fenofibrate action should be better specified.
Response 2: We agree with your comment. We have added the definition of metabolic syndrome as per the World Health Organization and detailed the mechanism of fenofibrate action with appropriate references.
["World Health Organization (WHO) defines metabolic syndrome as... microalbuminuria [3]."]
["Fenofibrate-mediated activation of PPAR-α leads to... apolipoproteins A-I and A-II synthesis [8–12]."]
3) Study design:
Comments: Dosage of fenofibrate is not clear.
Response 3: We apologize for the oversight. We have clarified that the only available fenofibrate formulation in the Russian Federation is the 145 mg film-coated tablets.
["The only available fenofibrate formulations in Russian Federation are 145 mg film-coated tablets."]
4) 3.3.2 Target subgroups:
Comments: Figures relating the results of the subgroups were not present.
Response 4: Thank you for bringing this to our attention. We have made significant revisions to this section and added figures to represent the changes in TG levels among main preplanned subgroups of interest.
5) Discussion:
Comments: Discussion and the Safety Section (4.2) should be modified.
Response 5: We have made substantial revisions to the discussion section and the safety section based on your feedback.
6) Language and Typing errors:
Comments: Extensive editing of the English language is required by an English native speaker.
Response 6: We have consulted an English language service provided by MDPI to refine the language and rectify the typographical errors in the manuscript. We appreciate your diligence in highlighting this.
7) References:
Comments: References should be updated.
Response 7: We have updated the references as per your recommendation.
Response to Comments on the Quality of English Language
Point 1:
Comments: Extensive editing of English language is required.
Response 1 : We recognize the importance of clear language in conveying our research. We have sought the services of a native English speaker to ensure our manuscript meets the required standards.
Additional clarifications:
We would like to extend our gratitude for the thorough review and valuable feedback. It has greatly enhanced the quality of our manuscript. We hope that the revisions meet the expectations and standards of the journal.
Warm regards,

Reviewer 2 Report
The manuscript is overall well written and the results are presented in a balanced way. No changes or corrections are needed in my opinion.
Author Response
Dear Reviewer,
We sincerely thank you for your thoughtful and positive feedback on our manuscript. Your recognition of the quality of our work and the clarity of its presentation is greatly appreciated. We are grateful for your meticulous review and the time you have invested in evaluating our paper. Your positive assessment motivates us and reaffirms our commitment to maintaining high standards in our research.
Thank you once again for your invaluable contribution to the peer-review process.
Warm regards,
Marat Ezhov and Gregory Arutyunov
Reviewer 3 Report
The revised manuscript, entitled “ Effectiveness of fenofibrate in routine treatment of hypertriglyceridemia” by Marat Ezhov and Gregory Arutyunov is generally well written but has low novelty. The fenofibrate treatment is well-known and is widely described in the literature. Please indicate the novelty of this study.
Other comments:
1. There is also lack of p values for differences on fig. 2 and 3
2. There is lack the results for subgroups ? You only wrote that “ TG changes in the prespecified subgroups were consistent with results from the over-all cohort.
3. The list of references is short (about half of older than 5 years old).
Author Response
Response to Reviewer Comments
Dear Reviewer,
Thank you for your valuable comments and constructive feedback on our manuscript titled “Effectiveness of fenofibrate in routine treatment of hypertriglyceridemia”. We greatly appreciate the time and effort you have dedicated to reviewing our work. Here are our responses to your comments:
Regarding the novelty of our study:
While it is true that fenofibrate treatment is well-known and widely described in the literature, it is important to underscore that there remains a significant gap in the literature concerning real-world evidence on the effectiveness and safety of fibrates. Our study aims to bridge this gap by providing fresh, real-world insights into the routine treatment of hypertriglyceridemia using fenofibrate. This makes our study particularly relevant for clinicians who are keen on understanding the practical applications and implications of fenofibrate treatment in everyday clinical settings. We have emphasized this aspect further by adding the statement: "It is also noteworthy that there is a particular lack of real-world evidence on fibrates effectiveness and safety [19]."
Other comments:
1. Lack of p values on fig. 2 and 3:
Response: Thank you for pointing this out. We have added the necessary information regarding p values. The updated figure notes now state: "dots represent mean value and the error bars are standard deviations; † – p < 0.0001 compared to baseline."
2. Lack of results for subgroups:
Response: We apologize for the oversight. Based on your valuable feedback, we have added detailed information about the subgroups, including corresponding graphs. We initially believed that adding these details might make the manuscript dense, but after considering your comment, we realized its importance. Thank you for guiding us on this aspect.
3. List of references:
Response: We appreciate your observation regarding our references. We have taken steps to update our reference list and bibliography to ensure that it reflects the most current and relevant research in this area.
Once again, we would like to express our gratitude for your insightful comments and suggestions. Your feedback has significantly contributed to improving the overall quality of our manuscript.
Warm regards,
Marat Ezhov
Gregory Arutyunov

Round 2
Reviewer 1 Report
Dear Authors,
all my comments and suggestions were satisfactorily met in the revised version of Your manuscript.